

# Exploration of key mechanisms underlying the therapeutic effects of AMD3100 on attenuating lipopolysaccharide-induced acute lung injury in mice

Zhou Lv[1,*], Bohan Zhang[1,*], Hui Zhang[1], Yanfei Mao[1], Qihong Yu[2] and Wenwen Dong[1]

[1] Department of Anesthesiology and Surgical Intensive Care Unit, Xinhua Hospital, Shanghai Jiaotong University School of Medicine, Shanghai, China
[2] Department of Gastroenterology, Changhai Hospital, The Second Military Medical University, Shanghai, China
* These authors contributed equally to this work.

## ABSTRACT

**Context:** AMD3100, a CXCR4 antagonist, has beneficial effects immaculate in the treatment of acute lung injury (ALI).

**Objective:** ALI is a severe inflammatory condition associated with poor prognosis and limited treatment options. AMD3100, has therapeutic effects that reduce ALI. Our study explored the regulatory mechanisms of AMD3100 in alleviating the injury of lipopolysaccharide (LPS)-induced ALI in mice.

**Materials and Methods:** Male ICR mice were randomly divided into control, LPS-treated, AMD3100-treated, and LPS + AMD3100-treatment groups. The histological changes of lung tissues from different groups were evaluated using hematoxylin and eosin staining. Lung injury was measured by ELISA and lung wet/dry ratio. Moreover, lung tissues from the four groups were subjected to transcriptome sequencing followed by differential expression, functional enrichment, protein-protein interaction (PPI) networks, and transcription factor analyses. The validation of mRNAs and protein levels were conducted with qRT-PCR and ELISA.

**Results:** Hematoxylin and eosin staining combined with the concentration of IL-1 and IL1-β and lung wet/dry ratios revealed that AMD3100 reduced the level of LPS-induced lung injury. Analysis of the transcriptome sequencing data identified 294 differentially expressed genes in the LPS-induced ALI mouse model. Based on the PPI network and module analysis, hub targets of AMD3100, such as Cxcl10 and Cxcl9, were identified in module 1, and hub targets, such as Cxcl12 and Cxcl1, were identified in module 2. Cxcl10 and Cxcl9 are involved in the Toll-like receptor signaling pathway, and Cxcl12 and Cxcl1 arae enriched in the nuclear factor-kappa B signaling pathway. Cxcl19, Cxcl10, and Cxcl1 are targeted by transcription factors like NF-κB. The validation of mRNAs and protein levels conducted by PCR and ELISA supported our transcriptome data.

**Conclusions:** Our findings indicate that AMD3100 may exhibit a therapeutic effect on LPS-induced ALI in mice by modulating multiple chemokines to inhibit the Toll-like receptor/nuclear factor-kappa B signaling pathway.

Corresponding authors
Qihong Yu, yuqihongaaaa@163.com
Wenwen Dong, dongwenwen@xinhuamed.com.cn

# INTRODUCTION

Acute lung injury (ALI) is a severe inflammatory syndrome which manifests as dyspnea and cyanosis due to insufficient oxygenation (*Guan et al., 2023*). It can be caused by various intra- and extra-pulmonary injury factors, such as ischemia-reperfusion, trauma, and sepsis (*Ning et al., 2023*). Acute respiratory distress syndrome (ARDS) is the acute onset of hypoxaemia and bilateral pulmonary oedema due to excessive alveolocapillary permeability according to the Berlin definition (*Meyer, Gattinoni & Calfee, 2021*). ALI plays a critical role in the development of acute respiratory distress syndrome (ARDS), frequently leading to catastrophic respiratory failure and high mortality rate (*Peritore et al., 2021*; *Mac Sweeney & McAuley, 2016*). While meticulous management of mechanical ventilation and judicious fluid therapy have been demonstrated to enhance patient outcomes significantly, the therapeutic options for ALI/ARDS remain starkly limited (*Matthay, McAuley & Ware, 2017*; *Szabari et al., 2019*). Due to the limited treatment options available and the severe nature of ALI/ARDS, high mortality rate still remains, ranging from 30% to 40% (*Matthay et al., 2019*). Consequently, it is imperative to to delve into the fundamental and crucial mechanisms underlying ALI and forge new, efficacious treatments.

The G protein-coupled receptor C-X-C motif chemokine receptor 4 (CXCR4) highly expressed in multiple tissues, such as the lung and vascular epithelial cells (*Ghosh et al., 2012*; *Regard, Sato & Coughlin, 2008*). The chemokine receptor CXCR4 plays a pivotal role in the activation of endothelial cells, promoting the accumulation of inflammatory cells at the site of inflammation through its interactions with the vascular endothelium (*Kantele, Kurk & Jutila, 2000*; *He et al., 2022*). Numerous studies have confirmed that CXCR4 has lung protective effects in multiple experimental models of ALI/ARDS (*Nassoiy et al., 2018*; *Hirano et al., 2018*) and may be a key drug target for treating ALI (*Lou et al., 2022*). AMD3100 is a CXCR4 antagonist that attenuates multiple pulmonary diseases (*Komolafe & Pacurari, 2022*). For instance, *Chen et al. (2016)* revealed that AMD3100 attenuates asthmatic responses, such as airway inflammation and responsiveness. *Li et al. (2020)* have demonstrated that AMD3100 ameliorates bleomycin-induced lung fibrosis in mice. *Drummond et al. (2015)* reported that the administration of AMD3100 has demonstrated an ability to diminish pulmonary inflammation and enhance both alveolar formation and the structural development of lung vasculature in neonatal rats experiencing lung injury induced by hyperoxia, according to evidence presented in reference. This evidence underscores the prospective utility of AMD3100 as a therapeutic agent in the context of ALI/ARDS. Despite the promising therapeutic implications of AMD3100 in the context of ALI/ARDS, the exact pathways and mechanisms by which AMD3100 exerts its protective effects against ALI/ARDS are not completely understood.

Lipopolysaccharide (LPS), an endotoxin found in the outer membrane of Gram-negative bacteria, acts as a potent stimulator for the release of inflammatory

cytokines and the generation of reactive oxygen species (*Lv et al., 2016*). Lipopolysaccharide (LPS) is a widely employed agent for inducing Acute Lung Injury (ALI) (*Zhang et al., 2021*; *Rahmawati et al., 2021*). In our investigation, we explored the impact of AMD3100 on LPS-induced ALI in mice by establishing a mouse model of LPS-induced ALI. Transcriptome sequencing and bioinformatic analysis were conducted to unravel the mechanism through which AMD3100 acts against ALI. The outcomes of this research provide novel insights into potential prophylactic and therapeutic strategies for Acute Lung Injury (ALI).

## MATERIALS AND METHODS

### Animals

Male ICR mice, aged 7–9 weeks and weighing 23–28 g, were procured from Shanghai Ji hui Laboratory Animal Care Co., Ltd. They were housed in groups of three in a specific pathogen-free environment at the Shanghai Xinhua Hospital animal laboratory, with a humidity range of 40–60% and a constant temperature of 24 ± 2 °C. The study comprised four experimental groups, where animals were initially weighed and subsequently numbered by weight before being randomly allocated to either the experimental groups or the control group ($n = 7$) using a randomization tool. No specific criteria were set for the inclusion or exclusion of animals in the study; all groups and animals were included in the analysis. The study was ethically approved by the Xinhua Hospital ethics committee (Approval ID: XHEC-F-2020-026, April 12th, 2020). The researchers adhered strictly to the principles of "Replacement, Reduction, and Refinement" for the welfare of experimental animals, implementing effective care measures to safeguard animal welfare and prevent unnecessary harm. The researchers possessed the requisite ability and qualifications for conducting the project, backed by clinical research experience and infrastructure. The approval letter is provided as Supplemental Material.

### Drug treatment

The animals were randomly assigned to four groups: control, LPS, AMD3100, and LPS plus AMD3100 (LPS + AMD3100), each comprising seven mice. Mice in the LPS group were intraperitoneally injected with LPS (5 mg/kg; Sigma-Aldrich, St. Louis, MO, USA). AMD3100 was administered intraperitoneally at a dosage of 10 mg/kg, one hour before LPS injection. The control group mice received an equivalent volume of saline. After a 24-hour period post-treatment, the mice were euthanized. Anesthesia was induced using 5% isoflurane in oxygen within a plexiglass cage, following which the mice were sacrificed, and lung tissues were harvested.

### Lung tissue histological evaluation

The lung tissues underwent fixation with 4% paraformaldehyde, followed by dehydration utilizing various concentrations of ethanol. These dehydrated tissues were then embedded in paraffin and sectioned into slices measuring 5 μm in thickness. Hematoxylin and eosin staining were performed on these sections. Slices are scanned and imaged under the microscope lens of a slice scanner (Pannoramic MIDI, Pannoramic 250 FLASH,

Pannoramic DESK; 3Dhistech, Budapest, Hungary), and seamlessly pieced together into a full field digital slice through a control software system (CaseViewer, C.V.2.4). The total surface of the slides was scored by two blinded pathologists with expertise in lung pathology. The criteria for scoring lung inflammation was set up as previously described with some modifications (*Wu et al., 2013*): 0, normal tissue; (1) minimal inflammatory change; (2) mild to moderate inflammatory changes (no obvious damage to the lung architecture); (3) moderate inflammatory injury (thickening of the alveolar septae); (4) moderate to severe inflammatory injury (formation of nodules or areas of pneumonitis that distorted the normal architecture); (5) severe inflammatory injury with total obliteration of the field.

## Enzyme-linked immunosorbent assay

ELISA was used to quantify the secretion of pro-inflammatory cytokines included IL-6, IL-1β in the BALF and Cxcl12, Cxcl10, Cxcl1 and Cxcl9 in the lung tissue. The lung tissue and BALF were collected after sacrifice of mice, stored at −80 °C and tested using ELISA kits.

## Transcriptome sequencing

The lung tissues from the four groups underwent transcriptome sequencing, and four samples were selected from each group. Total RNA was extracted from the lung tissues using the mirVana miRNA Isolation Kit (Ambion, Austin, TX, USA). After detecting RNA integrity, the libraries were established using the TruSeq Stranded mRNA LT Sample Prep Kit (Illumina, San Diego, CA, USA) and then sequenced on the HiSeq 2500 sequencing platform (Illumina, San Diego, CA, USA), resulting in the generation of paired-end reads with lengths of 125 bp/150 bp.

## Data filtering and mapping

The raw reads generated were processed using Trimmomatic (*Bolger, Lohse & Usadel, 2014*). After filtering out the reads containing poly-N and low-quality reads, clean reads were obtained and aligned to the reference genome using HISAT2 (*Kim, Langmead & Salzberg, 2015*). Cufflinks (*Trapnell et al., 2010*) were used to calculate the FPKM value, and htseq-count (*Anders, Pyl & Huber, 2015*) was applied to evaluate the read counts.

## Differential expression analysis

The FPKM values were converted into log2 (FPKM+1). The differentially expressed genes between the LPS and control groups, and between the LPS + AMD3100 and LPS groups, were identified using the limma package (version 3.10.3) (*Smyth, 2005*). The cutoff values were set at log fold change >2 and adjusted *p*-value <0.05. Genes with opposite expression trends in the LPS *vs.* control and LPS + AMD3100 *vs.* LPS groups were used as target genes for subsequent analyses.

## Functional enrichment analysis

To elucidate the function of the target genes, Gene Ontology (GO) (*Ashburner et al., 2000*) and Kyoto Encyclopaedia of Genes and Genomes (KEGG) (*Kanehisa & Goto, 2000*) pathway enrichment analyses were performed using cluster profile (version 3.18.1)

| Table 1 Gene primer sequence. | | |
| --- | --- | --- |
| GENE | Primer sequence | |
| Cxcl12 | Forward | TGCATCAGTGACGGTAAACCA |
| | Reverse | CACAGTTTGGAGTGTTGAGGAT |
| Cxcl10 | Forward | CCAAGTGCTGCCGTCATTTTC |
| | Reverse | GGCTCGCAGGGATGATTTCAA |
| Cxcl9 | Forward | GGAGTTCGAGGAACCCTAGTG |
| | Reverse | GGGATTTGTAGTGGATCGTGC |
| Cxcl1 | Forward | ACTGCACCCAAACCGAAGTC |
| | Reverse | TGGGGACACCTTTTAGCATCTT |

(*Yu et al., 2012*). The significant enrichment results were obtained with threshold values of count ≥2 and adjusted $p$-value < 0.05.

## Construction of protein-protein interaction network

The PPI relationships between proteins encoded by the target genes were predicted with the STRING (version 10.0) database (*Szklarczyk et al., 2015*). The parameters were set as follows: mouse as the organism, and a PPI score of 0.4. A PPI network was constructed with Cytoscape (version 3.2.0). The topological properties of each node were analyzed using the CytoNCA plugin (version 2.1.6) (*Tang et al., 2015*). The hub nodes in the PPI network were screened by ranking the scores of each node. Additionally, significant modules in the network were selected using the MCODE plugin (version 1.4.2) (*Bandettini et al., 2012*). The threshold value was greater than five. Functional enrichment analyses of the module genes were also conducted.

## Prediction of transcription factor

Transcription factors (TFs) potentially targeting module genes were anticipated through the utilization of the Web Gestalt GAST tool (*Zhang, Kirov & Snoddy, 2005*). Key parameters included the selection of *Mus musculus* as the Organism of Interest, the application of Overrepresentation Enrichment Analysis as the Method of Interest, a significance level set at $p < 0.05$, and the focus on the Top 10 outcomes.

## Quantitative RT-PCR for the validations of mRNA

To validate the RNA sequencing data, quantitative real-time polymerase chain reaction (qRT-PCR) was employed, revealing significant differences in four mRNAs across the four groups ($n = 7$ per group). Lung tissue samples were utilized for sequencing, with all gene primers detailed in Table 1. Initially, total RNA was extracted from the lung using the RNA iso Plus Kit (TaKaRa, Shiga, Japan). Subsequently, the PrimeScript™ RT reagent Kit (TaKaRa, Shiga, Japan) facilitated the reverse transcription of RNA into cDNA. qRT-PCR assays were performed using the SYBR® Premix Ex Taq™ kit (TaKaRa, Shiga, Japan) on the biosystems QuantStudio 5 Flex (Thermo Fisher Scientific, Waltham, MA, USA) as per the manufacturer's instructions. The relative expression levels of mRNAs in the lung were

normalized to 18S, while the relative mRNA expression was normalized to U6. The $2^{-\Delta\Delta Ct}$ method was utilized for calculating the qRT-PCR results, with each qRT-PCR sample undergoing three replications.

## Statistical analysis

All data were analyzed with the statistical program SPSS 25.0 (Chicago, IL, USA). Data are expressed as means ± SEM. Statistical comparisons between experimental groups and control group were performed using two sample t-test (T-T test) with an additional Bonferroni *post hoc* test. Multiple groups were compared using one-way ANOVA test followed by Bonferroni correction. Statistical significance was defined as *p*-value < 0.05.

## RESULTS

### AMD3100 alleviated LPS-induced ALI in mice

The protective effect of AMD3100 was assessed using Hematoxylin and eosin staining. Normal pulmonary structures were observed in the lung tissues in control group. After LPS treatment, the lung tissues showed evident histopathological alterations, such as alveolar wall edema and infiltration of inflammatory cell. However, after further treatment with AMD3100, the alveoli in the LPS + AMD3100 group exhibited remarkable restoration in morphology, accompanied by minor pathological changes (Fig. 1A). We evaluated the histopathological damage scores of the different groups. The histopathological damage score exhibited a marked increase in the LPS group compared with the control group (*p* < 0.01). Furthermore, compared with the LPS group, the histopathological damage score had remarkably decreased in the LPS + AMD3100 group (*p* < 0.01) (Fig. 1B). We also measured the lung wet/dry ratio in different groups, which indicated the damage level of lung. The LPS group had an increased ratio compared with the control group. However, the lung wet/dry ratio significantly decreased in the LPS + AMD3100 group (*p* < 0.01) (Fig. 1C). The concentration of IL-6 and IL-1β in BALF indicated that the LPS group had an increased level of inflammation, which showed a decreased tend in the LPS + AMD3100 group (Fig. 1D). These findings indicate that AMD3100 alleviated LPS-induced acute lung injury of mice.

### Transcriptome sequencing data

In total, 113.37 G of clean data were obtained from the sequencing of 16 samples. The effective data size for each sample ranged from 6.75 to 7.38 Gb, with Q30 base distribution ranging from 95.4% to 95.71% and an average GC content of 50.55% (Table 2). The alignment of clean reads to the reference genome had alignment rates ranging from 96.7% to 97.88% (Table 3).

### Identification of target genes

Using the limma package, 541 differentially expressed genes (291 upregulated and 250 downregulated) were identified in the LPS *vs.* control group (Fig. 2A), and 307 genes (131 upregulated and 276 downregulated) were identified in the LPS + AMD3100 *vs.* LPS group

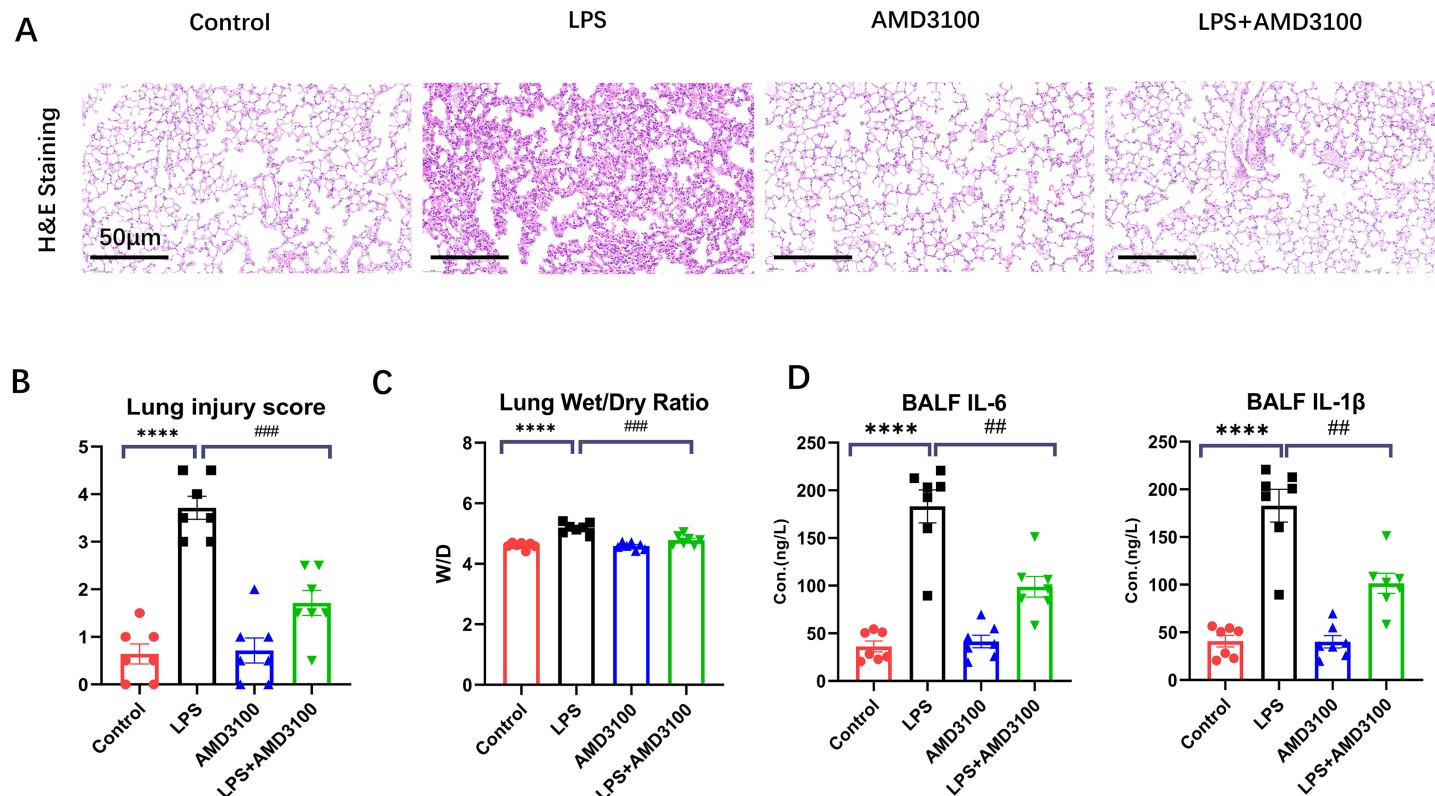

**Figure 1  AMD3100 alleviated LPS-induced ALI in mice.** (A) Pathological images of control, LPS, AMD3100, and LPS + AMD3100 group. (B) Lung injury scores of the different groups. (C) Lung wet/dry ratio of the different groups. (D) The concentration of IL-6 and IL1-β in BALF. The data are expressed as mean ± SEM ($n$ = 7 in each group). ****$p$ < 0.0001 compared to control group. ##$p$ < 0.01, ###$p$ < 0.001, compared with the LPS group. ALI, acute lung injury; LPS, lipopolysaccharide; SEM, standard error of the mean; H&E, hematoxylin and eosin staining.

(Fig. 2B). Venn analysis yielded 172 intersecting genes from the upregulated genes in the LPS *vs*. control group and downregulated genes in the LPS + AMD3100 *vs*. LPS group, alongside 122 intersecting genes from the downregulated genes in the LPS *vs*. control group and upregulated genes in the LPS + AMD3100 *vs*. LPS group (Fig. 3). These 294 intersecting genes were selected as target genes for subsequent analyses.

## Functional enrichment analysis for target genes

To elucidate the functions of the target genes, we analyzed the enriched GO terms and KEGG pathways. These target genes were enriched in 551 GO biological process functions, such as the cytokine-mediated signaling pathway, response to the virus, and response to interferon-beta (Fig. 4A). Moreover, nine pathways were significantly enriched; these included cytokine-cytokine receptor interaction, nuclear factor-kappa B (NF-κB) signaling pathway, TNF signaling pathway, and IL-17 signaling pathway (Fig. 4B).

## PPI network analysis and hub gene identification

A PPI network was constructed with the target genes, which included 219 nodes and 1,023 interactions (Fig. 5). Hub nodes were identified based on topology scores. The top five

**Table 2 The quality preprocessing results of sequencing data.**

| Sample | Raw reads | Raw bases | Clean reads | Clean bases | Valid bases (%) | Q30 (%) | GC (%) |
|---|---|---|---|---|---|---|---|
| A1 | 51.68M | 7.75G | 50.77M | 7.36G | 94.99 | 95.45 | 50.56 |
| A2 | 47.32M | 7.10G | 46.55M | 6.75G | 95.14 | 95.61 | 50.55 |
| A3 | 49.18M | 7.38G | 48.31M | 6.98G | 94.67 | 95.53 | 50.58 |
| A4 | 50.71M | 7.61G | 49.89M | 7.26G | 95.45 | 95.59 | 50.63 |
| B1 | 50.81M | 7.62G | 50.07M | 7.28G | 95.54 | 95.71 | 50.41 |
| B2 | 47.84M | 7.18G | 47.03M | 6.79G | 94.69 | 95.66 | 50.42 |
| B3 | 49.75M | 7.46G | 48.94M | 7.12G | 95.36 | 95.58 | 50.47 |
| B4 | 47.77M | 7.17G | 47.02M | 6.84G | 95.41 | 95.59 | 50.38 |
| C1 | 48.71M | 7.31G | 47.92M | 7.00G | 95.78 | 95.40 | 50.51 |
| C2 | 50.53M | 7.58G | 49.73M | 7.25G | 95.62 | 95.45 | 50.63 |
| C3 | 47.49M | 7.12G | 46.74M | 6.81G | 95.56 | 95.48 | 50.58 |
| C4 | 49.09M | 7.36G | 48.35M | 7.05G | 95.81 | 95.56 | 50.40 |
| D1 | 51.49M | 7.72G | 50.70M | 7.38G | 95.58 | 95.61 | 50.97 |
| D2 | 48.85M | 7.33G | 48.13M | 7.02G | 95.78 | 95.67 | 50.41 |
| D3 | 50.84M | 7.63G | 50.02M | 7.30G | 95.76 | 95.50 | 50.69 |
| D4 | 49.92M | 7.49G | 49.17M | 7.18G | 95.84 | 95.52 | 50.54 |

**Table 3 The sequence alignment results.**

| Sample | Total reads | Total mapped reads | Multiple mapped | Uniquely mapped | Reads mapped in proper pairs |
|---|---|---|---|---|---|
| A1 | 50,766,214 | 49,156,500 (96.83%) | 2,280,465 (4.49%) | 46,876,035 (92.34%) | 45,843,372 (90.30%) |
| A2 | 46,546,504 | 45,538,486 (97.83%) | 2,336,844 (5.02%) | 43,201,642 (92.81%) | 42,254,164 (90.78%) |
| A3 | 48,312,504 | 47,231,242 (97.76%) | 2,315,223 (4.79%) | 44,916,019 (92.97%) | 43,925,758 (90.92%) |
| A4 | 49,892,192 | 48,785,113 (97.78%) | 2,424,642 (4.86%) | 46,360,471 (92.92%) | 45,378,354 (90.95%) |
| B1 | 50,070,298 | 48,794,515 (97.45%) | 2,439,041 (4.87%) | 46,355,474 (92.58%) | 45,282,068 (90.44%) |
| B2 | 47,026,778 | 45,801,828 (97.40%) | 2,383,940 (5.07%) | 43,417,888 (92.33%) | 42,335,186 (90.02%) |
| B3 | 48,944,648 | 47,705,368 (97.47%) | 2,352,061 (4.81%) | 45,353,307 (92.66%) | 44,280,588 (90.47%) |
| B4 | 47,021,142 | 45,808,787 (97.42%) | 2,320,541 (4.94%) | 43,488,246 (92.49%) | 42,466,164 (90.31%) |
| C1 | 47,916,624 | 46,334,625 (96.70%) | 2,243,561 (4.68%) | 44,091,064 (92.02%) | 43,144,824 (90.04%) |
| C2 | 49,728,658 | 48,666,831 (97.86%) | 2,427,556 (4.88%) | 46,239,275 (92.98%) | 45,222,388 (90.94%) |
| C3 | 46,735,166 | 45,727,789 (97.84%) | 2,102,670 (4.50%) | 43,625,119 (93.35%) | 42,665,788 (91.29%) |
| C4 | 48,345,436 | 47,321,138 (97.88%) | 2,473,394 (5.12%) | 44,847,744 (92.77%) | 43,895,872 (90.80%) |
| D1 | 50,700,630 | 49,525,576 (97.68%) | 2,311,604 (4.56%) | 47,213,972 (93.12%) | 46,137,922 (91.00%) |
| D2 | 48,129,460 | 47,064,360 (97.79%) | 2,412,489 (5.01%) | 44,651,871 (92.77%) | 43,664,660 (90.72%) |
| D3 | 50,023,998 | 48,914,220 (97.78%) | 2,416,502 (4.83%) | 46,497,718 (92.95%) | 45,468,210 (90.89%) |
| D4 | 49,165,792 | 48,075,893 (97.78%) | 2,216,470 (4.51%) | 45,859,423 (93.28%) | 44,836,940 (91.20%) |

nodes with the highest degrees were the C-X-C motif chemokine ligand 10 (Cxcl10, degree = 57), interferon regulatory factor 7 (Irf7, degree = 49), Cxcl9 (degree = 48), Irf1 (degree = 45), and cluster of differentiation 274 (Cd274, degree = 41).

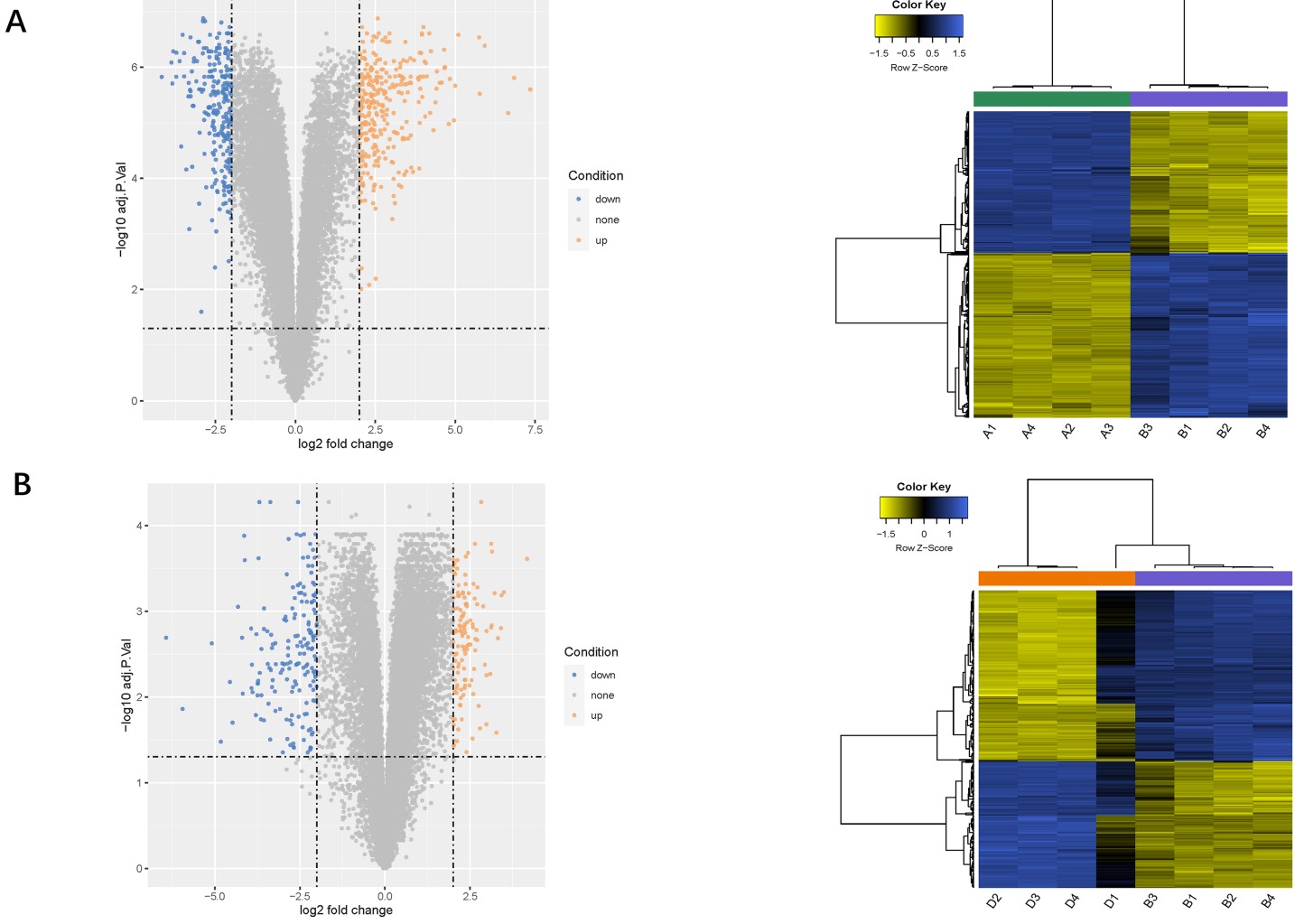

**Figure 2 Analysis of DEGs.** (A) Volcano plot and heat map of DEGs between the LPS (group B) and control group (group A). (B) Volcano plot and heat map of DEGs between the LPS + AMD3100 (group D) and LPS group (group B) (*n* = 4 in each group). DEGs, differentially expressed genes; LPS, lipopolysaccharide.

## Module analysis and functional enrichment analysis

Using the MCODE plugin (score > 5), two modules were identified in the PPI network (score > 5; Fig. 6A). Module 1 (score = 28.207) contained 30 nodes (such as Cxcl10 and Cxcl9) and 409 interaction pairs. Module 2 (score = 6.667) contained 10 nodes (such as Cxcl12 and Cxcl1) and 30 interaction pairs.

Moreover, the genes in module 1 were enriched in 133 GP-BP terms, such as response to the virus, and 11 KEGG pathways, such as the Toll-like receptor (TLR) and TNF signaling pathways (Fig. 6B). Notably, Cxcl10 and Cxcl9 are involved in the TLR signaling pathway. The genes in module 2 were enriched in 188 GP-BP terms, such as cell chemotaxis, and eight KEGG pathways, such as cytokine-cytokine receptor interaction and NF-κB signaling pathway (Fig. 6C). Remarkably, Cxcl12 and Cxcl1 were involved in cytokine-cytokine receptor interaction and the NF-κB signaling pathway.
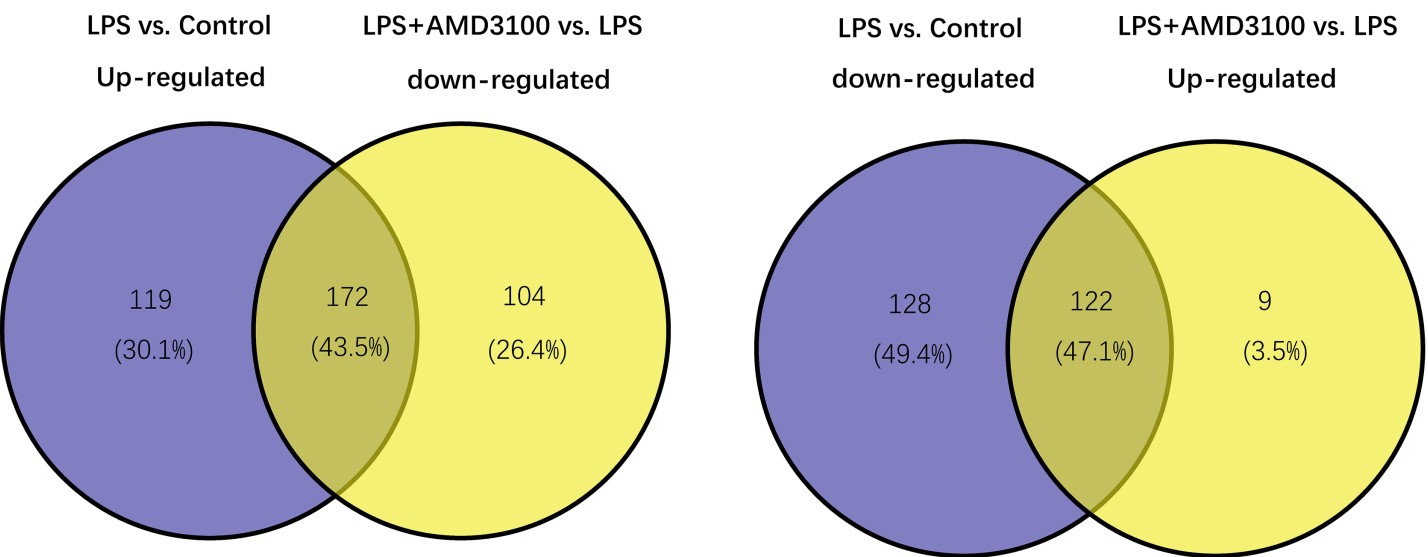

**Figure 3 Venn diagram analysis of target genes of AMD3100 in LPS-induced ALI.** ALI, acute lung injury; LPS, lipopolysaccharide.

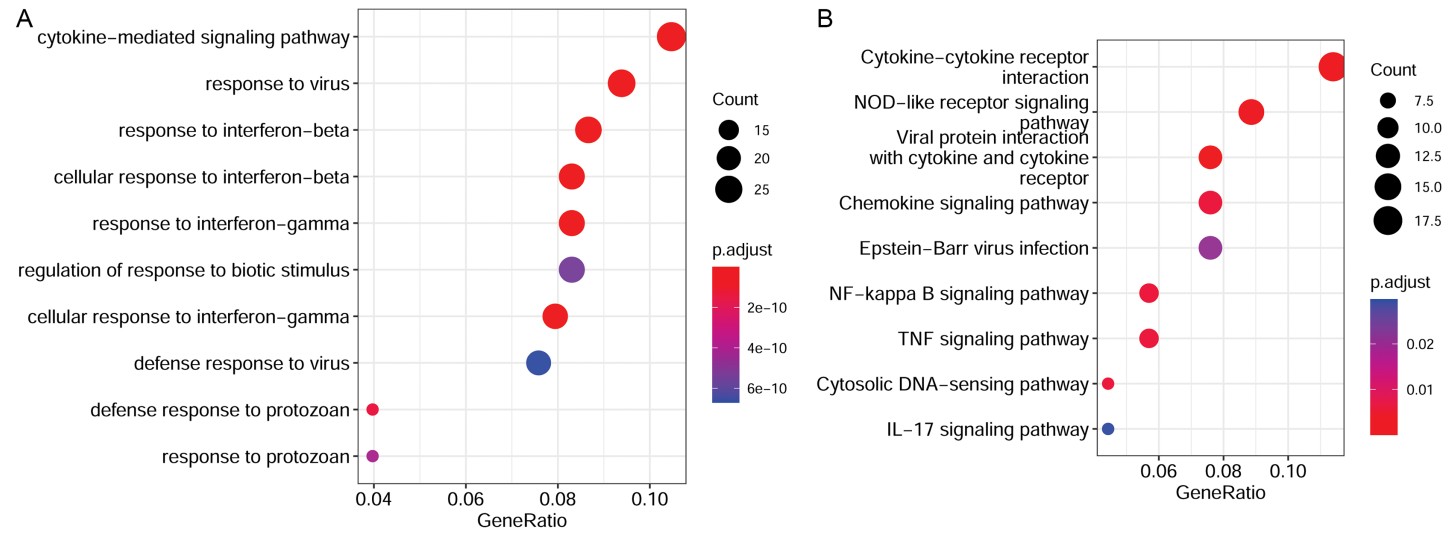

**Figure 4 Functional enrichment results of target genes.** (A) Top 10 significantly enriched GO biological process functions. (B) Top 10 significantly enriched KEGG pathways. GO, Gene Ontology; KEGG, Kyoto Encyclopaedia of Genes and Genomes.

## TF-target regulatory network analysis

Using the Web Gestalt GAST tool, the top ten TFs that could target module genes were obtained. A TF-target regulatory network was constructed, which included 10 TFs, 15 module genes, and 49 TF-target relationships (Fig. 7). Modular genes in this network were downregulated. Moreover, Cxcl10, Cxcl9, and Cxcl1 were targeted by NF-κB (NFKAPPAB_01 and NFKB_Q6).

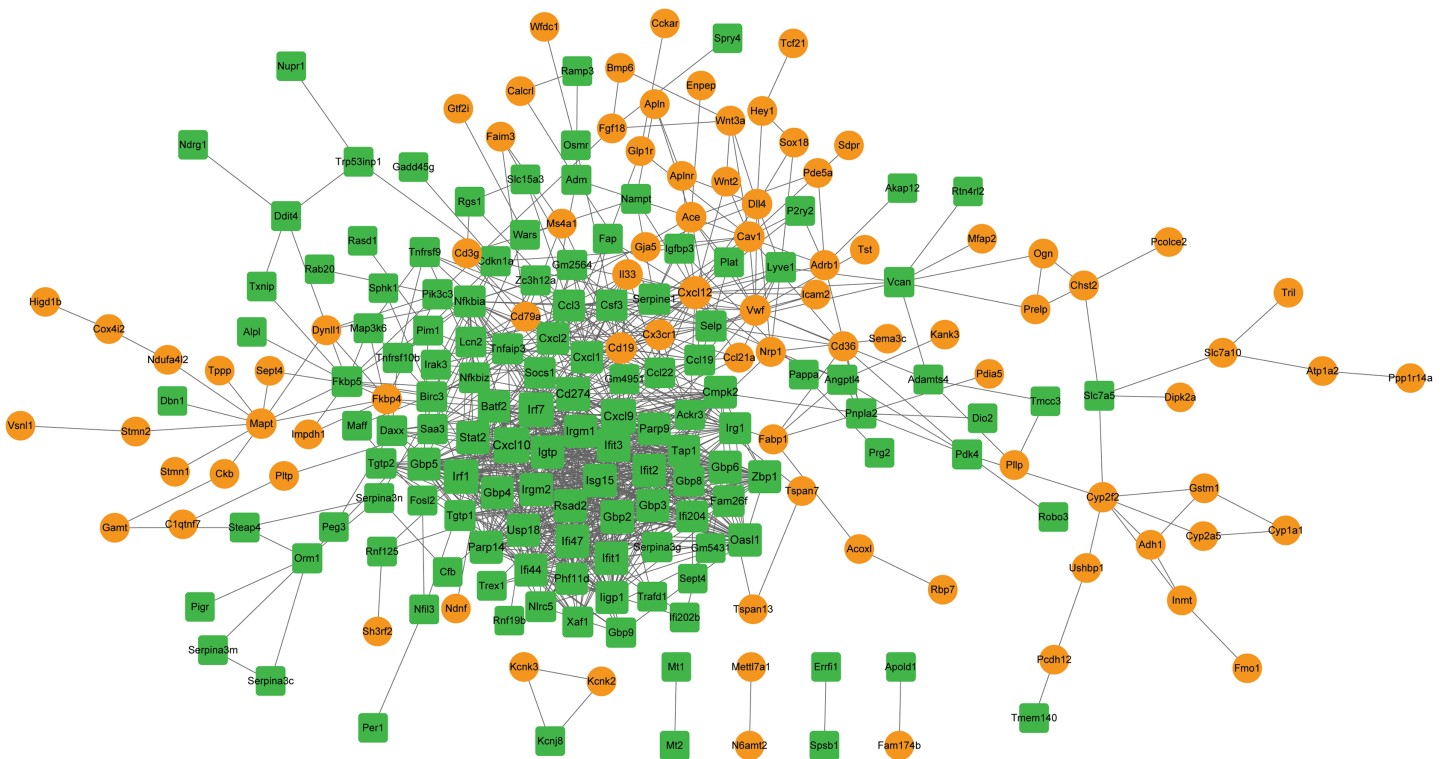

**Figure 5 The PPI network is constructed by target genes.** Yellow circle nodes indicate upregulated proteins between the LPS + AMD3100 and LPS groups, and green squares represent downregulated proteins between the LPS + AMD3100 and LPS groups. The size of the node depends on its degree. The higher the degree value, the larger the node. PPI, protein-protein interaction; LPS, lipopolysaccharide.

## Validations of mRNA and protein

In order to verify the DEGs, we selected four significantly changed mRNAs. We performed qRT-PCR from the lung tissues for the validation of mRNA, which was used for mRNA sequencing. The expressions of Cxcl12, Cxcl10, Cxcl9, and Cxcl1 were increased in the LPS group compared with that in the control group and decreased in the LPS + AMD group using 18s for endogenous control (Fig. 8A). We also measured the concentration of Cxcl12, Cxcl10, Cxcl9, and Cxcl1 from the lung tissues with ELISA. The expressions of Cxcl12, Cxcl10, Cxcl9, and Cxcl1 were increased in the LPS group compared with that in the control group and decreased in the LPS + AMD group (Fig. 8B). The results of qRT-PCR and ELISA are significant support to demonstrate our research.

## DISCUSSION

Acute lung injury is a challenging inflammatory disorder with limited prognosis and treatment options (*Fanelli & Ranieri, 2015*). Previous studies have shown that AMD3100, a CXCR4 antagonist, can alleviate LPS-induced ALI (*Yaxin et al., 2014*; *Liu et al., 2023*). However, the regulatory mechanisms by which AMD3100 attenuates inflammatory lung injury remain unknown. Present study illustrated that AMD3100 markedly alleviated LPS-induced ALI in mice. Transcriptome sequencing identified 294 target genes of AMD3100 in the mouse model of LPS-induced ALI. Through PPI network and module

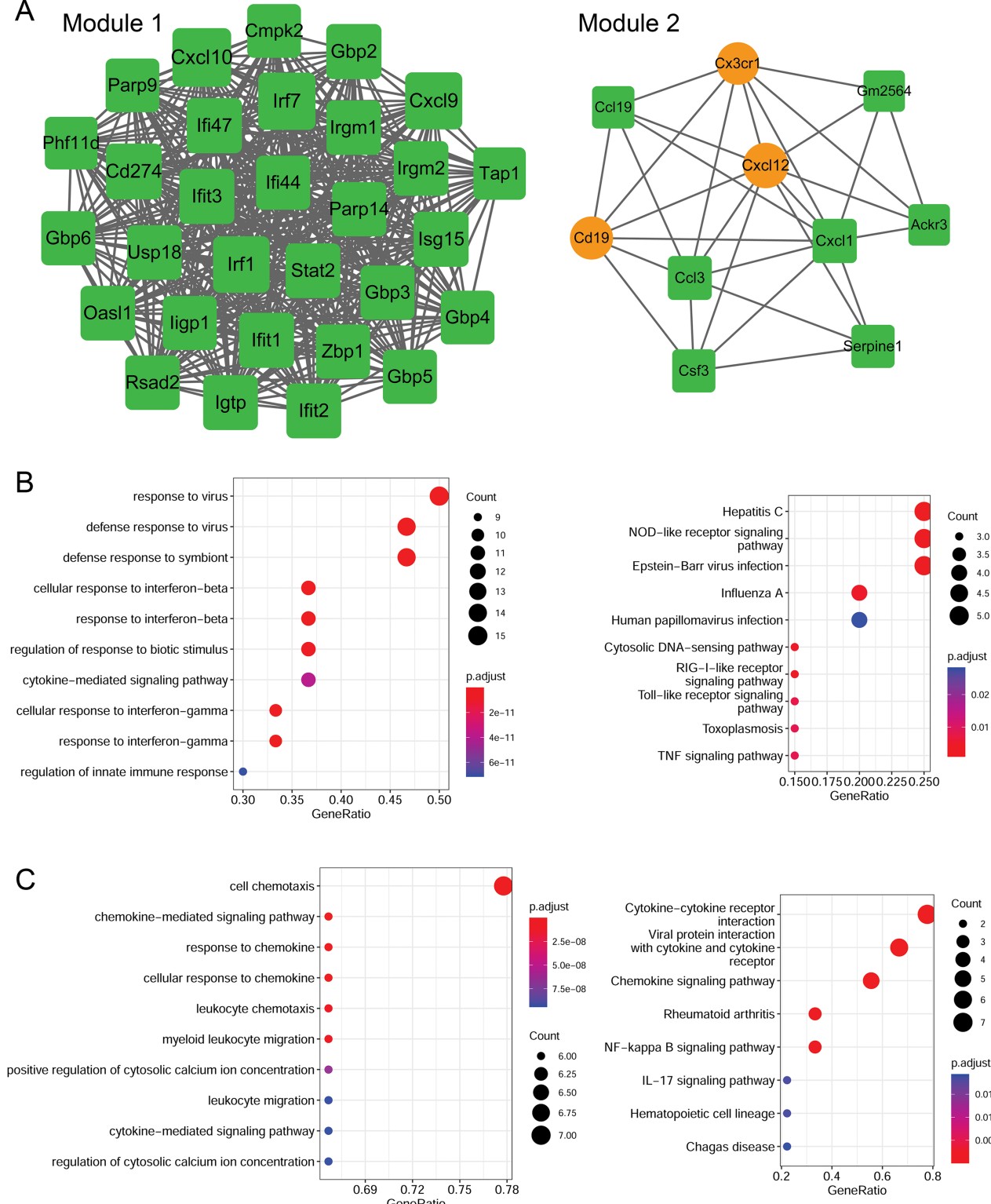

**Figure 6 Identification of PPI modules and functional enrichment results of module genes.** (A) PPI modules. (B) GO and KEGG enrichment results for module 1. (C) GO and KEGG enrichment results for module 2. Yellow circle nodes indicate upregulated proteins between the LPS + AMD3100 and LPS groups, and green squares represent downregulated proteins between the LPS + AMD3100 and LPS groups. The size of the node depends on its degree. The higher the degree value, the larger the node. PPI, protein-protein interaction; LPS, lipopolysaccharide; GO, Gene Ontology; KEGG, Kyoto Encyclopaedia of Genes and Genomes.

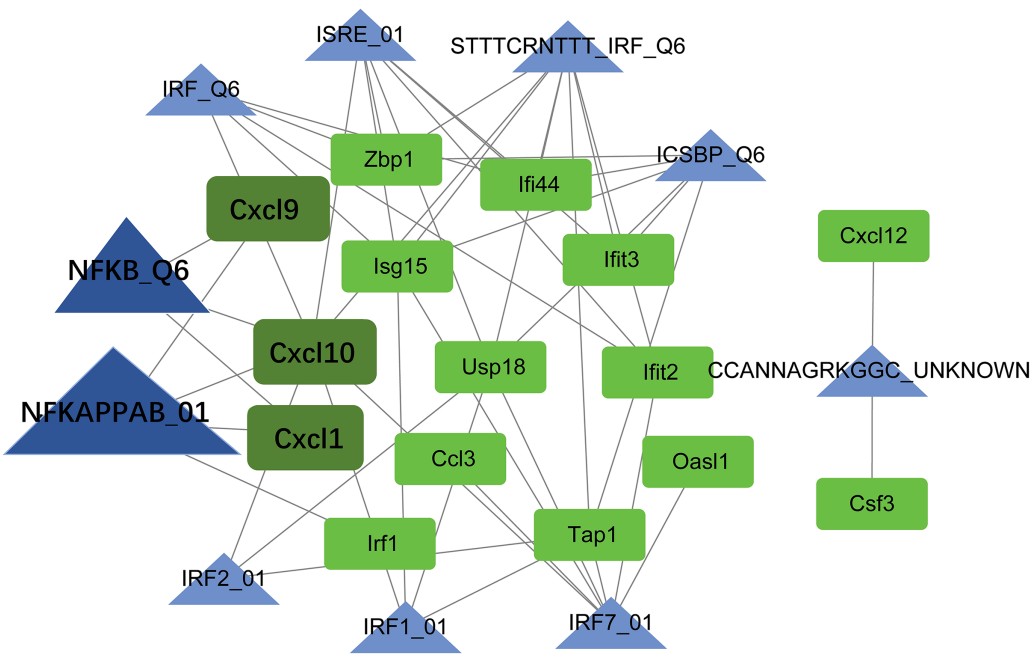

**Figure 7 The TF-target regulatory network.** The green square node indicates the module gene, and the blue triangle node indicates TF. The higher the degree value, the larger the node. TF, transcription factor.

analysis, hub targets of AMD3100, such as Cxcl10 and Cxcl9 were identified in module 1, and Cxcl12 and Cxcl1 were identified in module 2. Moreover, Cxcl10 and Cxcl9 were involved in the TLR signaling pathway, while Cxcl12 and Cxcl1 were enriched in NF-κB signaling pathway. Furthermore, Cxcl10, Cxcl9, and Cxcl1 could be targeted by TFs, such as NF-κB. These findings reveal the potential mechanism of action of AMD3100 in LPS-induced ALI.

ALI is marked by an acute inflammation within the pulmonary tissues Chemokines play a pivotal role in the onset and development of inflammatory reactions (*Sahin & Wasmuth, 2013*). In our study, multiple chemokines, such as Cxcl10, Cxcl9, and Cxcl1, were identified as hub targets of AMD3100 and were markedly downregulated in the LPS + AMD3100 group relative to the LPS group. The chemokine Cxcl10, recognized as Interferon-γ inducible protein 10, orchestrates immune activity by guiding inflammatory cells to areas of inflammation (*Scolletta et al., 2013*). Cxcl10 is widely considered an essential inflammatory mediator in various inflammation-associated diseases, such as ulcerative colitis (*Mostafa & Abdel-Rahman, 2023*), non-alcoholic steatohepatitis, pulmonary tuberculosis (*Ali, Mankhi & Ad'hiah, 2021*), and non-alcoholic steatohepatitis (*Zhang et al., 2014*). A previous study suggested that Cxcl10 is a promising biomarker for ARDS (*Xie et al., 2019*). The expression of chemokine Cxcl9 is also induced by interferon-γ in a Type I immune response and the molecule acts as a chemoattractant for mononuclear cells. Cxcl9 participates in ALI by modulating the inflammatory and immune response (*Zheng et al., 2022*). Cxcl1 acts as a chemoattractant for neutrophils and has been

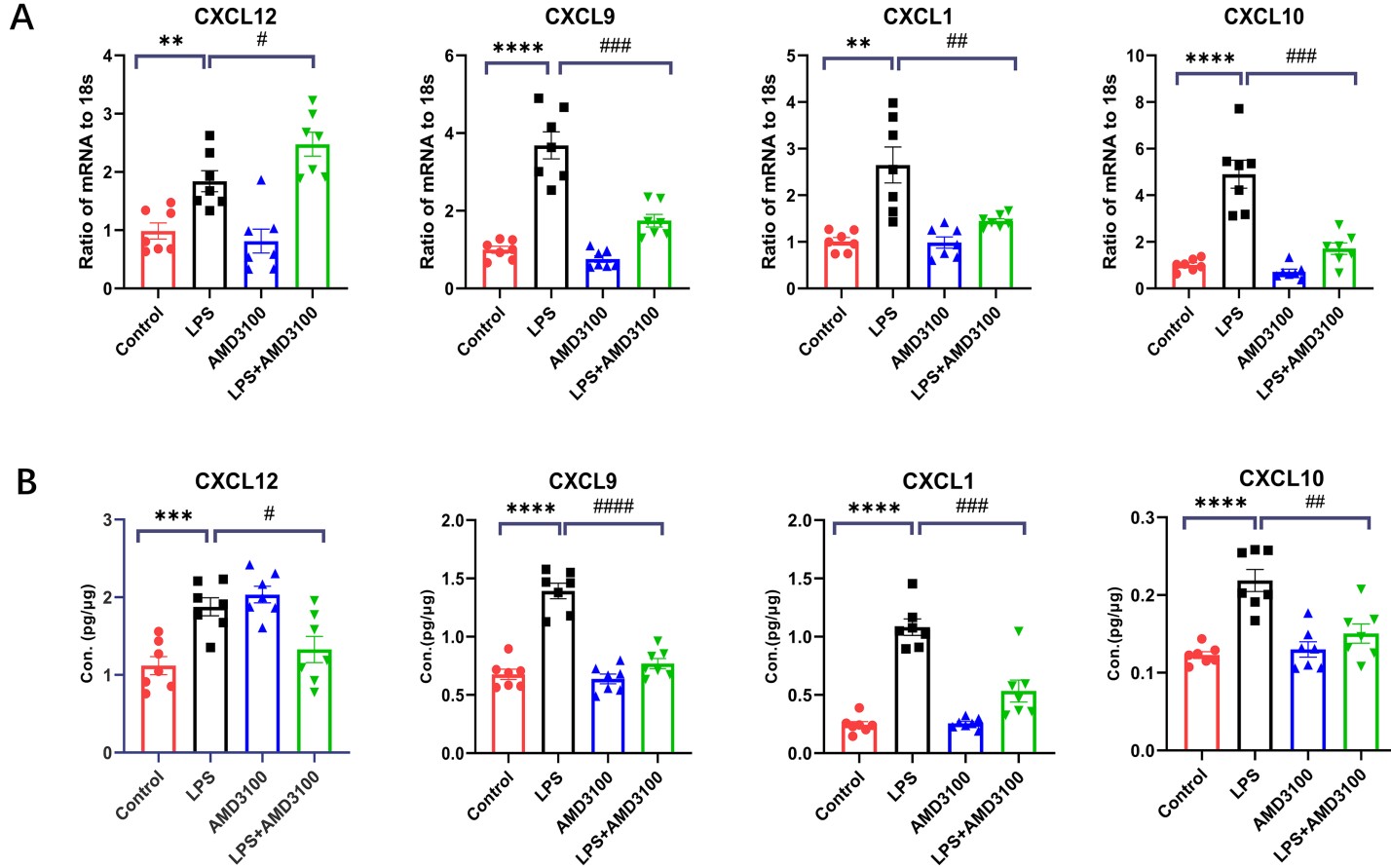

**Figure 8 The verification of key genes.** (A) The relative values of Cxcl12, Cxcl 9, Cxcl 1, Cxcl10 in four groups conducted with qRT-PCR. (B) The concentration of Cxcl12, Cxcl9, Cxcl1, Cxcl10 in lung tissue conducted with ELISA. The data are expressed as mean ± SEM ($n = 7$ in each group). **$p < 0.01$, ***$p < 0.001$, ****$p < 0.0001$ compared to control group. #$p < 0.05$, ##$p < 0.01$, ###$p < 0.001$, ####$p < 0.0001$ compared with the LPS group. SEM, standard error of the mean.

implicated in neutrophil influx during lung inflammation (*Yee et al., 2021*). Neutrophils recruited by Cxcl1 to infected tissue play a crucial role in eliminating pathogens during ALI (*Meng et al., 2018*). These findings reveal that AMD3100 may attenuate LPS-induced ALI by regulating dysregulated chemokines.

Additionally, Cxcl2 was identified as a hub target of AMD3100. Cxcl12 is produced by pulmonary epithelial cells in the lungs, which can attract leukocytes to the airspaces during ALI (*Petty et al., 2007*). Accumulating evidence has confirmed that Cxcl12 is upregulated in ALI/ARDS and may result in excessive recruitment of neutrophils into pulmonary tissues, which in turn can initiate inflammation, cause tissue injury, and ultimately lead to fibrotic changes (*Isles et al., 2019*; *He et al., 2022*). Cxcl12 is also associated with pulmonary inflammation, indicating its involvement in inflammatory processes that occur in the lungs (*Gao et al., 2018*; *Chen et al., 2015*). Notably, Cxcl12 is the only chemokine ligand of CXCR4, and numerous studies have shown that CXCL12 recruits inflammatory cells in a CXCR4-dependent manner (*Tettamanti et al., 2006*; *Tao, Yuan & Liao, 2016*). The Cxcl12/

CXCR4 axis was implicated in neutrophilic inflammation (*Mousavi, 2020*). Moreover, AMD3100 interferes with the interaction of CXCR4 and its ligand, Cxcl12, thus inhibiting its activity (*Vorobjeva & Pinegin, 2014*; *Sercundes & Ortolan, 2016*). The findings indicate that Cxcl12 could be a critical mediator in the inflammatory response associated with ALI through its interaction with the CXCR4 receptor. Although our study demonstrated an increase in Cxcl12 expression within the LPS + AMD3100 group, we speculate that the obstructed interaction between Cxcl12 and its receptor CXCR4, facilitated by AMD3100, Cxcl12, might impede the functional activity of Cxcl12.

Multiple pathways were enriched to elucidate the regulatory mechanisms of AMD3100 in ALI. For example, we found that Cxcl10 and Cxcl9 are involved in the TLR signaling pathway. In previous studies, Cxcl10 was linked to lung injury and activation of the TLR4 signaling pathway (*Imai et al., 2008*; *Barrenschee, Lex & Uhlig, 2010*). Activation of the TLR signaling pathway is implicated in the progression of inflammatory responses in ALI (*Wu et al., 2018*). Additionally, Cxcl10, Cxcl9, and Cxcl1 could be targeted by NF-κB, and Cxcl12 and Cxcl1 were enriched in the NF-κB signaling pathway. The NF-κB is a critical TF located downstream of the TLR signaling pathway. Multiple studies have substantiated the involvement of NF-κB in the pathogenesis of LPS-induced ALI (*Wang et al., 2014*; *Yang, Li & Chen, 2021*). During sepsis, the presence of LPS can trigger the TLR4 signaling pathway, leading to the translocation of NF-κB, subsequently impacting the expression of pro-inflammatory cytokines, such as IL-1β and interleukin-6 (IL-6) (*Jiang et al., 2015*). *Zou et al. (2023)* demonstrated that goniothalamin prevented LPS-induced ALI and inflammation *via* the TLR4/NF-κB signaling pathway. Collectively, the TLR/NF-κB signaling pathway may represent a crucial mechanism through which AMD3100 delivers its protective impact against LPS-induced ALI.

In conclusion, our data preliminarily reveals that AMD3100 may alleviate LPS-induced ALI by modulating multiple chemokines to suppress the TLR/NF-κB signaling pathway. These findings provide a scientific foundation for ALI management.

### Funding

This work was supported by the National Natural Science Foundation of China (No. 82070662). The funders had no role in study design, data collection and analysis, decision to publish, or preparation of the manuscript.

### Grant Disclosures

The following grant information was disclosed by the authors:
National Natural Science Foundation of China: 82070662.

### Competing Interests

The authors declare that they have no competing interests.

## Author Contributions

- Zhou Lv conceived and designed the experiments, authored or reviewed drafts of the article, and approved the final draft.
- Bohan Zhang analyzed the data, prepared figures and/or tables, and approved the final draft.
- Hui Zhang analyzed the data, prepared figures and/or tables, and approved the final draft.
- Yanfei Mao performed the experiments, prepared figures and/or tables, and approved the final draft.
- Qihong Yu performed the experiments, authored or reviewed drafts of the article, and approved the final draft.
- Wenwen Dong conceived and designed the experiments, authored or reviewed drafts of the article, and approved the final draft.

## Animal Ethics

The following information was supplied relating to ethical approvals (*i.e.*, approving body and any reference numbers):

Xinhua Hospital Ethics Committee Affiliated to Shanghai Jiaotong University School of Medicine provided full approval for this research.

## DNA Deposition

The following information was supplied regarding the deposition of DNA sequences:

The data is available at NCBI SRA: SRR28351781, SRR28351780, SRR28351779, SRR28351778, SRR28351777, SRR28351776, SRR28351775, SRR28351774, SRR28351773, SRR28351772, SRR28351771, SRR28351770, SRR28351769, SRR28351768, SRR28351767, SRR28351766; PRJNA1085512; SAMN40354775.

## Data Availability

The raw measurements are available in the Supplemental File.

## Supplemental Information

Supplemental information for this article can be found online at http://dx.doi.org/10.7717/peerj.18698#supplemental-information.

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
