# Peer review of "Exploration of key mechanisms underlying the therapeutic effects of AMD3100 on attenuating lipopolysaccharide-induced acute lung injury in mice"

_PeerJ, doi:10.7717/peerj.18698_

## Round 0.1 · original submission · Major Revisions

Requirements for improvements provided by all three reviewers are very serious but I hope that the authors will be able to respond properly.

Reviewer 1 ·

Basic reporting

The authors report AMD3100, a CXCR4 antagonist, alleviates acute lung injury as a prophylactic measure.

Major concerns
1. No verification experiment has been conducted for transcriptome sequencing.
2. What is the rationale behind administering AMD3100 treatment 1 hour before LPS injection as a prophylactic measure?

Minor concerns
1. Figure 1: Images of HE lack a scale bar.
2. Figure 2A: To which group do A and B belong? Figure 2B: To which group do D and B belong?
3. Figure 3: The picture shown was incomplete.
4. The name of each group is not consistent in bioinformatic analysis. Please verify the name of each group. Please double-check the name of each group.
5. Based on what was the GO-KEGG analysis conducted?

Experimental design

What is the rationale behind administering AMD3100 treatment 1 hour before LPS injection as a prophylactic measure?

Validity of the findings

No verification experiment has been conducted for transcriptome sequencing.

Reviewer 2 ·

Basic reporting

The manuscript satisfies most of the journal's requirements. There are no further comments here.

Experimental design

The study's logic is well presented. However, there are some small concerns from the method section:
1. How did the authors analyze the H&E images? What software was used, and was the analysis conducted double- or single-blind to prevent potential bias?
2. Measuring the lung wet/dry ratio is more reliable than relying solely on H&E staining to assess the severity of lung injury.
3. No molecular mechanism is investigated.

Validity of the findings

There are some concerns raised from the study:
1. No scalebar was added to the H&E figures in Fig.1; please consider adding it.
2. Please consider using a bar graph with scatter plots to present Fig. 1B.
3. Please add the sample size (n =?) to the figure legend.
4. In Fig. 3, the left part of the panel, which is LPS+AMD3100 vs LPS (upregulated), is incomplete.

·

Basic reporting

The English are fine, with the text presenting a few typos.
The background should include LPA and ARDS definitions because, since 2013, the therm ALI has no longer been used according to the Berlin Definition. However, many authors still misuse it.
The structure, figure, table, and data are OK. However, some new data must be included. For instance, the dosage of Cxcl10, Cxcl9, Cxcl10, IL-1B, and Il-6 should be included in the BALF to support the discussion and improve the manuscript.

Experimental design

The study sounds scientific, with straightforward questions and a good study design. However, it would benefit from more experimental data, such as the dosage of Cxcl10, Cxcl9, Cxcl10, IL-1B, and Il-6 from the BALF or lung tissue.
Different signs in the histological images should indicate the tissue alterations and a full lung image should be provided. Further details on the "lung score" in MM are needed. In addition, include in all figure legends the number of animals/analyses used in each group.
Figure 7 highlights the five nodes.

Validity of the findings

The study includes new data in the field, and the discussion and conclusion are OK if new experimental data are provided.

---

## Round 0.2 · Minor Revisions

Let me positively interpret responses of the reviewers as satisfaction with the introduced changes. However, the Section Editor (Dr. Brenda Oppert) brought to my attention to the fact that "at present the English of the manuscript sometimes is not conveying the intended meaning." You should edit the entire manuscript to improve English of it. It should be done in line with the suggestions that Dr. Oppert made for the abstract:

Context: AMD3100, a CXCR4 antagonist, possesses beneficial pathological effects inmaculate lung injury (ALI).

Dr. Oppert thinks that the authors mean to say "AMD3100, a CXCR4 antagonist, has beneficial effects immaculate in the treatment of acute lung injury (ALI)."

Editing the remainder of the abstract:

Objective: ALI is a severe inflammatory condition associated with poor prognosis and limited treatment options. AMD3100, has therapeutic effects that reduce ALI. Our study explored the regulatory mechanisms of AMD3100 in alleviating the injury of lipopolysaccharide (LPS)-induced ALI in mice.
Materials and Methods: Male ICR mice were randomly divided into control, LPS-treated, AMD3100-treated, and LPS + AMD3100-treatment groups. The histological changes of lung tissues from different groups were evaluated using hematoxylin and eosin staining. Lung injury was measured by ELISA and lung wet/dry ratio. Moreover, lung tissues from the four groups were subjected to transcriptome sequencing followed by differential expression, functional enrichment, protein-protein interaction (PPI) networks, and transcription factor analyses. The validation of mRNAs and protein levels were conducted with qRT-PCR and ELISA.
Results: Hematoxylin and eosin staining combined with the concentration of IL-1and IL1-β and lung wet/dry ratios revealed that AMD3100 reduced the level of LPS-induced lung injury. Analysis of the transcriptome sequencing data identified 294 differentially expressed genes in the LPS-induced ALI mouse model. Based on the PPI network and module analysis, hub targets of AMD3100, such as Cxcl10 and Cxcl9, were identified in module 1, and hub targets, such as Cxcl12 and Cxcl1, were identified in module 2. Cxcl10 and Cxcl9 are involved in the Toll-like receptor signaling pathway, and Cxcl12 and Cxcl1 arae enriched in the nuclear factor-kappa B signaling pathway. Cxcl19, Cxcl10, and Cxcl1 are targeted by transcription factors like NF-κB. The validation of mRNAs and protein levels conducted by PCR and ELISA supported our transcriptome data.
Conclusions:
Our findings indicate that AMD3100 may exhibit a therapeutic effect on LPS-induced ALI in mice by modulating multiple chemokines to inhibit the Toll-like receptor/nuclear factor-kappa B signaling pathway."

·

Basic reporting

The authors answered the questions raised.

Experimental design

OK

Validity of the findings

OK

Additional comments

The authors answered the questions raised.

---

## Round 0.3 · accepted · Accept

Thank you for your efforts. I am positive that the commentaries of the reviewers contributed to improvement of the manuscript.